# Association Between Genomic Features and Radiation Response in Metastatic Breast Cancer Patients Undergoing Palliative Radiotherapy

**DOI:** 10.3390/ijms262411837

**Published:** 2025-12-08

**Authors:** Hyeon Seok Choi, Sejoon Lee, So Yeon Park, Jee Hyun Kim, Se Hyun Kim, Koung Jin Suh, Seung Hyuck Jeon, In Ah Kim

**Affiliations:** 1Department of Radiation Oncology, Seoul National University College of Medicine, Seoul 03080, Republic of Korea; lastson92@snu.ac.kr; 2Precision Medicine Center, Seoul National University Bundang Hospital, Seongnam 13620, Republic of Korea; sejoonlee@snu.ac.kr; 3Department of Pathology, Seoul National University Bundang Hospital, Seoul National University College of Medicine, Seongnam 13620, Republic of Korea; sypmd@snu.ac.kr; 4Department of Internal Medicine, Seoul National University Bundang Hospital, Seoul National University College of Medicine, Seongnam 13620, Republic of Korea; jhkimmd@snu.ac.kr (J.H.K.); sehyunkim@snubh.org (S.H.K.); skjmd0919@snubh.org (K.J.S.); 5Department of Radiation Oncology, Seoul National University Bundang Hospital, Seongnam 13620, Republic of Korea; hyck9004@snubh.org

**Keywords:** metastatic breast cancer, palliative radiotherapy, biologically effective dose, next-generation sequencing, PI3K–Akt signaling pathway, radiosensitivity biomarkers

## Abstract

Responses to palliative radiotherapy (RT) for metastatic lesions vary among patients, and molecular determinants of radiosensitivity remain unclear. This study investigated genomic features associated with local progression-free survival (LPFS) in metastatic breast cancer patients treated with palliative RT. Forty-four patients who underwent next-generation sequencing of 523 cancer-related genes were retrospectively analyzed. The biologically effective dose (BED) was calculated using an α/β ratio of 3 Gy, and local progression was defined as recurrence or progression within the irradiated field. A total of 60 metastatic lesions, predominantly in bone (68.3%), were evaluated. Higher BED (≥88 Gy) was significantly associated with longer LPFS (*p* = 0.011). Among 320 detected mutations mapped to 141 Kyoto Encyclopedia of Genes and Genomes (KEGG) pathways, and the PI3K–Akt signaling pathway remained an independent predictor in multivariate analysis (*p* = 0.03). Subgroup analyses demonstrated that patients with Ras, PI3K–Akt, or FoxO pathway mutations derived greater LPFS benefit from high BED, whereas this advantage was confined to wild-type tumors for the PD-L1/PD-1 checkpoint and choline metabolism pathways. These findings suggest that pathway-specific molecular contexts modulate RT response and may inform individualized radiation dose strategies in metastatic breast cancer.

## 1. Introduction

Breast cancer remains the most prevalent malignancy among women worldwide. Owing to advances in early detection and therapeutic strategies, survival rates have steadily improved. Notably, the survival of patients with metastatic breast cancer (MBC) has been prolonged through the development of systemic drug therapies. In hormone receptor-positive breast cancer, the addition of CDK4/6 inhibitors to endocrine therapy has demonstrated a significant improvement in progression-free survival (PFS) compared with that after endocrine monotherapy [1]. Similarly, in HER2-positive breast cancer, T-DM1 use has been associated with extended PFS [2].

Despite these advancements in systemic treatment, MBC remains an incurable disease, and patients often experience prolonged disease courses. During this period, most patients undergo palliative radiotherapy (RT) for symptom management and local tumor control. However, the therapeutic response to RT varies considerably among individuals with MBC. To date, predictive biomarkers for the RT response or toxicity have not been comprehensively identified or validated. Therefore, the implementation of personalized RT based on individual genetic or molecular profiles remains limited in clinical practice.

The advancement of next-generation sequencing (NGS) technology has significantly transformed clinical practice and accelerated the adoption of molecular targeted therapies. NGS is now widely employed during both the diagnostic and treatment planning stages, particularly in cases of tumor recurrence. Although NGS can be conducted for all patients with solid tumors depending on institutional protocols, in breast cancer, it is most utilized in the recurrent or metastatic setting to identify actionable mutations. Because genetic heterogeneity is a key driver of therapeutic resistance, accurate identification of genomic alterations is essential for guiding effective treatment strategies.

Growing experimental evidence indicates that specific genetic alterations may influence the tumor response to RT. However, to date, no genetic biomarkers have been validated for routine clinical use in predicting RT outcomes. Notably, somatic mutations in ATM have been associated with exceptional responses to RT across multiple tumor types, including head and neck, prostate, breast, and lung cancers [3]. Additionally, PIK3CA mutations have been linked to a reduced risk of local recurrence in breast cancer, and a potential role of the PI3K pathway in modulating radiosensitivity has been reported [4]. These findings highlight the need for further research to establish genomic predictors of the RT response.

In this study, we analyzed the genomic features of breast cancer in relation to local progression-free survival (LPFS) using NGS-based tumor panel testing in patients undergoing palliative RT on metastatic lesions.

## 2. Results

A total of 44 patients with MBC were included, encompassing 60 metastatic lesions. Patient characteristics are summarized in Table 1. The median age was 54.2 years old and ductal histology was the most common subtype (75.0%); the proportion of luminal A, luminal B, and triple-negative subtypes was 25%, 47.7%, and 27.3%, respectively. Bone was the most frequent site of metastasis (68.3%). Palliative RT was delivered to 60 metastatic lesions with a median BED of 75.0 Gy.

A total of 320 unique mutated genes were identified, representing distinct genes harboring mutations in at least one patient. Across the cohort, 791 mutation events were observed, which included multiple distinct mutations within the same gene in individual patients as well as recurrent mutations in the same gene across different patients. At the pathway level, KEGG pathway over-representation analysis performed using the enrichKEGG function revealed 141 pathway alterations, with a total of 1086 pathway alteration events identified across all patients. The detailed results are summarized in Appendix A.

When clinical variables were analyzed, the BED was significantly associated with LPFS both in univariate (Hazard ratio (HR) 0.98, 95% confidence interval (CI) 0.97–1.00; *p* = 0.020) and multivariate Cox analyses (HR 0.98, 95% CI 0.96–1.00; *p* = 0.020). This association remained significant after accounting for within-patient correlation using clustered robust standard errors (HR 0.98, 95% CI 0.96–1.00; *p* = 0.020). Using the maximally selected rank statistics method, a BED cutoff of 88 Gy was identified. After stratifying lesions into BED-high (*n* = 16) and BED-low (*n* = 44) groups, there was a significant difference in LPFS between the two groups (*p* = 0.01) (Figure 1).

Univariate Cox proportional hazards analysis was performed for 56 genes with a mutation frequency of at least 10%. Among these, five genes demonstrated a potential association with LPFS at a significance threshold of *p* < 0.1, including PIK3CA (HR, 0.40; 95% CI, 0.17–0.95; *p* = 0.037), NTRK3 (HR, 2.54; 95% CI, 1.02–6.33; *p* = 0.045), ERBB3 (HR, 2.42; 95% CI, 0.89–6.55; *p* = 0.083), BARD1 (HR, 0.17; 95% CI, 0.02–1.27; *p* = 0.085), and MYCN (HR, 0.29; 95% CI, 0.07–1.25; *p* = 0.097). Variables for the multivariate Cox regression model were selected by including gene mutations that met the threshold of *p* < 0.1 in univariate analysis. Among clinical factors, BED, estrogen receptor status, and Ki-67 index were included based on their significance in univariate analysis. The multivariate Cox model, including these five gene and clinical variables, identified ERBB3 mutation as an independent predictor of shorter LPFS (HR, 3.42; 95% CI, 1.12–10.46; *p* = 0.03; Table 2), and this association remained significant after accounting for within-patient clustering.

The frequency of the 141 identified pathway mutations ranged from 1.67% to 56.67%. Univariate Cox analysis for LPFS was performed on the 81 pathways with a mutation frequency ≥10%. The full results are presented in Appendix A. Twenty-five pathways demonstrated statistical significance (*p* < 0.05) in the univariate analysis. From the pathways with a *p*-value < 0.1 in the univariate analysis, those associated with other cancers or unrelated organ systems were excluded. A multivariate Cox analysis was then performed including the remaining 12 pathways—PI3K–Akt signaling pathway, Ras signaling pathway, EGFR tyrosine kinase inhibitor resistance, ErbB signaling pathway, FoxO signaling pathway, HIF-1 signaling pathway, VEGF signaling pathway, B cell receptor signaling pathway, PD-L1 expression and PD-1 checkpoint pathway in cancer, Choline metabolism in cancer, Estrogen signaling pathway, and Notch signaling pathway—along with BED, estrogen receptor status, and Ki-67 index as clinical covariates. Among these pathways, mutation in the PI3K–Akt signaling pathway was identified as an independent predictor of improved LPFS (HR, 0.18; 95% CI, 0.04–0.87; *p* = 0.03, Table 3). Because the number of events was small relative to the number of variables, clustered Cox regression could not be performed for this model.

Subgroup analysis based on KEGG pathway status showed that the effect of BED on LPFS differed by molecular context. In patients with mutations in the Ras, PI3K–Akt, and FoxO signaling pathways, a high BED was associated with significantly improved LPFS compared with that with a low BED (log-rank *p* = 0.04, 0.04, and 0.005, respectively), while no significant difference was observed in the corresponding wild-type groups (Figure 2). In contrast, for the PD-L1 expression and PD-1 checkpoint pathway in cancer and choline metabolism in cancer KEGG pathways, BED-related LPFS differences were observed only in wild-type tumors (*p* = 0.009 and 0.01, respectively), with no such differences in mutation-positive groups (Figure 3).

## 3. Discussion

In this study, we analyzed the relationship between the BED and LPFS in patients with MBC, integrating clinical and genomic variables, including gene- and pathway-level mutations. Our findings demonstrated that a high BED was associated with improved LPFS overall, and this association was particularly significant in patients with specific molecular alterations. The impact of BED on LPFS varied depending on pathway mutation status, emphasizing the importance of molecular context in modulating radiosensitivity. Notably, mutations in Ras and PI3K–Akt signaling pathway were associated with improved LPFS outcomes.

Clinically, the identified BED cutoff of 88 Gy holds significant biological relevance in the context of breast cancer. Widely used palliative regimens, such as 30 Gy in 10 fractions (BED = 60 Gy) or 20 Gy in 5 fractions (BED = 46.7 Gy), deliver doses well below this threshold and are primarily intended for symptom relief rather than durable tumor control. In contrast, achieving a BED of 88 Gy corresponds to approximately 50–54 Gy in conventional 2-Gy fractions, a dose range often required for sustained local control in metastatic lesions. High-dose ablative regimens in extracranial oligometastatic breast cancer typically deliver BED values in the range of 80–110 Gy and achieve 2-year local control approaching 90%, consistent with our observation that a BED cutoff around 88 Gy marks a biologically effective dose range for durable lesion control [5]. This suggests that the 88 Gy cutoff may represent a critical transition point where radiation shifts from purely palliative intent toward a biologically effective range capable of overcoming intrinsic or mutation-driven radioresistance. Given that pathway-level alterations in Ras, PI3K–Akt, and FoxO were associated with enhanced benefit from high BED in our study, this threshold likely reflects the dose intensity necessary to counteract the survival and DNA repair advantages conferred by these oncogenic signaling networks.

Although ERBB3 mutation emerged as a significant gene-level predictor, we aimed to further delineate the broader biological context underlying radiosensitivity. To capture functional relationships among multiple co-occurring mutations, we additionally performed pathway-level analysis using KEGG annotations. This approach allowed us to explore whether mutations converging on common signaling networks, such as PI3K–Akt or Ras pathways, could collectively modulate the radiation response, thereby providing a more comprehensive understanding beyond single-gene associations.

A methodological distinction is that our findings predominantly reflect upstream genomic alterations rather than the state of downstream effector activity [6,7]. Although the TruSight Oncology 500 panel includes key downstream MAPK effectors such as MAPK1 (ERK2) and MAPK3 (ERK1), no pathogenic mutations in these genes were detected in our cohort. Other MAPK-related genes covered by the panel (e.g., MAP2K2, MAP3K1) also exhibited mutation frequencies well below the threshold required for reliable survival analysis. As a result, the BED–LPFS association observed in this study is primarily driven by upstream alterations within the RAS or PI3K/AKT pathways rather than mutational events in downstream MAPK effectors. It should also be noted that functional MAPK activation is largely regulated by post-translational processes, including ERK phosphorylation, which cannot be assessed through DNA-based sequencing alone. Accordingly, our findings highlight the prognostic relevance of upstream drivers, while the contribution of downstream effector activation warrants further investigation using proteomic or phospho-signaling approaches.

However, several limitations should be considered. This was a retrospective, single-institution study with a relatively small sample size, which may limit the generalizability of the findings. The limited sample size relative to the genomic variables analyzed resulted in model instability and wide confidence intervals. Similarly, the optimal BED cutoff of 88 Gy was derived post hoc and may be unstable due to the small sample size. In addition, the analysis was conducted on a lesion-level basis, but inter-lesional heterogeneity and treatment-related factors, such as systemic therapy during RT, were not uniformly controlled. Specifically, 8 lesions received concurrent CDK4/6 inhibitors and 2 lesions received mTOR inhibitors. Given these small numbers, a reliable subgroup analysis isolating their effect or interaction with genomic alterations was not feasible. Because these agents target signaling networks closely related to pathways evaluated in this study, including the PI3K–Akt–mTOR axis and immune regulatory pathways, their concurrent use represents a potential confounder; however, their limited proportion suggests a modest influence on the overall cohort signal. Genomic profiling was performed using a targeted panel, which may not capture all relevant alterations, and the biopsy timing relative to RT could not be standardized. Frequent spatial discordance and a median 154-day interval between biopsy and radiotherapy limit the direct correlation due to potential clonal evolution or heterogeneity. Furthermore, the functional consequences of the identified mutations were not differentiated; both gain-of-function and loss-of-function mutations were grouped together in the analysis, complicating the biological interpretation of their associations with treatment outcomes. Because of the limited sample size and number of progression events, construction of a fully adjusted model incorporating potential confounders such as lesion site, fractionation regimen, concurrent systemic therapy, prior irradiation, and target volume was not feasible.

Before conducting the pathway-level survival analyses, we examined whether the presence of gene- or pathway-level alterations was associated with available clinical covariates, including estrogen receptor status, Ki-67 index, molecular subtype, histology, and metastatic site. Using Fisher’s exact test for categorical variables and the Wilcoxon rank-sum test for Ki-67, with the Benjamini–Hochberg false discovery rate correction applied within each covariate, no statistically significant associations were identified (q < 0.10). These findings suggest that the distribution of genomic alterations was largely independent of conventional clinicopathologic factors in this cohort, supporting that the observed associations between BED and LPFS were unlikely to be confounded by baseline clinical characteristics.

Of particular interest is the observation that the BED-related benefit from RT was more pronounced in tumors harboring mutations in the Ras, PI3K–Akt, and FoxO signaling pathways. This finding suggests that the presence of mutations in these pathways may influence the radiosensitivity of tumors and potentially confer resistance to RT.

Many studies have demonstrated increased radiation resistance in cells transfected with oncogenic Ras. Since Ras oncogene expression enhanced radiation survival in many transformed cells, studies were undertaken to test the hypothesis that inhibition of Ras expression or signaling would specifically radiosensitize tumor cells with activated Ras and showed inhibition of Ras radiosensitized various cancer cells (reviewed in [8]).

The PI3K–Akt signaling pathway is the key downstream pathway of Ras and plays a central role in regulating cell growth, proliferation, and survival and is implicated in processes such as apoptosis inhibition, tumor progression, metastasis, and radioresistance [9]. Preclinical studies have shown that selective inhibition of PI3K and Akt isoforms led to reduced Akt phosphorylation and enhanced radiosensitivity [10]. These findings support the notion that targeting PI3K–Akt signaling may radiosensitize tumors. Furthermore, recent studies suggest that PI3K inhibitors can also influence the tumor immune microenvironment, expanding their therapeutic potential beyond direct tumor cytotoxicity. Inhibition of PI3Kγ signaling can promote pro-inflammatory polarization of tumor-associated macrophages and enhance CD8+ T cell activation by activating NFκB and inhibiting C/EBPβ activation [11]. Moreover, combining PI3Kα/δ inhibitors with RT and PD-1 blockade can reduce tumor growth and enhance an abscopal effect [12]. PI3Kγ/δ inhibition with PD-1 blockade in combination with RT can enhance antitumor efficacy by reducing immunosuppressive cells and increasing CD8+ T cells [13].

Ras and FoxO signaling pathways are closely interconnected with the PI3K–Akt axis. Mutations in these pathways may lead to similar downstream effects on cell survival and DNA damage repair, thereby contributing to comparable modulation of the RT response [8]. Additionally, mutations in the Akt pathway are known to inactivate FoxO transcription factors, which are crucial for regulating apoptosis and DNA repair. This disruption reduces tumor-suppressive functions of FoxO, enhancing cell survival and promoting radioresistance by impairing apoptotic signaling and efficient DNA damage repair [14]. Therefore, the observed BED-specific differences in RT outcomes thus provide indirect evidence of the influence of oncogenic pathway mutations on intrinsic tumor radiosensitivity.

DNA double-strand breaks induced by RT can activate the ATM/ATR/Chk1 kinase cascade, which subsequently enhances PD-L1 expression via STAT1/3 and IRF1 signaling [15]. This increased PD-L1 functions as a critical immune evasion mechanism by suppressing cytotoxic T-cell activity. Supporting this finding, elevated PD-L1 expressions have been observed in rectal cancer patients with poor responses to neoadjuvant chemoradiotherapy, suggesting its role in radioresistance [16]. Therefore, we can speculate that mutations within this pathway may disrupt the immune evasion mechanism and potentially increase radiosensitivity via enhancing radiation-induced immunologic cell death, as observed in the present study. Similarly, alterations in the choline metabolism pathway can be interpreted within this framework. Hyperactivation of this pathway leads to elevated metabolite and enzyme levels [17]. These, in turn, activate downstream signaling cascades such as Ras and PI3K–Akt, promoting tumor progression and radioresistance [18]. Conversely, loss-of-function mutations in the choline metabolism pathway suppress these oncogenic signals and enhance tumor radiosensitivity.

The integration of genomic information, such as signaling pathway mutations, into radiation oncology practice represents a critical step toward precision medicine. One approach is the genome-based model for adjusting radiotherapy dose (GARD), which utilizes a radiosensitivity index derived from the expression of 10 key genes associated with the radiation response [19]. Although the GARD has shown predictive value in clinical datasets [20,21], its reliance on a limited gene panel presents inherent limitations. Comprehensive genomic profiling and further validation studies are necessary to refine biological dose modeling for practical use of personalized radiation dose prescription in clinics.

## 4. Materials and Methods

This was a single-institution retrospective study of patients diagnosed with breast cancer who received care at our center and underwent NGS-based genomic profiling between March 2020 and April 2022. Patients were excluded if they had non-metastatic disease or did not receive palliative RT for metastatic lesions. Additional exclusion criteria included the absence of pre-treatment imaging with computed tomography or magnetic resonance imaging of the irradiated site or lack of post-RT imaging follow-up to assess the treatment response. This study was approved by the institutional review boards of Seoul National University Bundang Hospital (IRB No. B-2102-667-110).

Genomic profiling was performed using the TruSight Oncology 500 (TSO500) kit (Illumina, San Diego, CA, USA) on DNA and RNA extracted from formalin-fixed paraffin-embedded (FFPE) tumor tissues, following the manufacturer’s hybrid capture-based library preparation protocol. The TSO500 panel includes 1.94 Mb of DNA targets covering 523 cancer-related genes and 358 kb of RNA targets spanning 55 genes for the detection of RNA fusions and splice variants.

Somatic variants were analyzed using the TSO500 Local App V2.2 pipeline, with filtering thresholds set at a variant allele frequency (VAF) ≥2% and a population frequency <1% in East Asian databases (gnomAD, 1000 Genomes). Copy number variations (CNVs) were reported when the fold change was ≥2.2. Gene fusions were identified using the RNA-based MANTA-Fusion algorithm, requiring at least three supporting reads. Tumor mutational burden (TMB) was calculated using only regions with ≥50× coverage within the effective panel size (1.33 Mb). For subsequent analyses, only single-nucleotide variants and small insertions/deletions were included.

Pathway-level alterations were identified using the Kyoto Encyclopedia of Genes and Genomes (KEGG) pathway over-representation analysis, implemented via the enrichKEGG function of the clusterProfiler package in R. The analysis was performed based on the observed gene mutations in each case. Specifically, enrichKEGG employs a hypergeometric statistical model to determine whether the set of mutated genes in a given sample is significantly enriched in predefined KEGG pathways compared to a background gene set. Pathway significance was assessed using adjusted *p*-values to account for multiple comparisons.

Local progression was defined as radiologic evidence of disease progression within the irradiated field, characterized by either the emergence of new lesions or a measurable increase in the size of previously identified lesions on follow-up imaging studies. The biologically effective dose (BED) was calculated using the linear-quadratic model with an assumed α/β ratio of 3 Gy. Univariate and multivariate Cox proportional hazards regression analyses were performed to evaluate the associations of clinical variables, the BED, individual gene mutations, and KEGG pathway alterations with LPFS. All statistical analyses were conducted on a lesion-level basis, with each irradiated metastatic lesion considered as a distinct analytical unit. Lesion-level Cox models were refitted with patient-clustered robust standard errors (clustered by the patient identifier) to account for within-patient correlation. Genes with a mutation frequency below 10% were excluded to reduce variance and overfitting from rare events. The maximally selected rank statistics method was used to determine the optimal BED cutoff. Survival differences between groups were assessed using the log-rank test. A *p*-value < 0.05 was considered statistically significant. All statistical analyses were performed using R version 4.2.1 (http://www.r-project.org, accessed on 28 october 2025).

## 5. Conclusions

Taken together, this study showed that the therapeutic efficacy of RT in MBC is significantly influenced by the BED and underlying genomic alterations of tumor. Specifically, mutations in the Ras, PI3K–Akt, and FoxO signaling pathways were associated with enhanced local control in response to a high BED. In contrast, alterations in immune regulatory pathways, such as PD-1/PD-L1 signaling, and metabolic pathways, such as choline metabolism, appeared to mitigate BED-related benefits. These findings suggest that pathway-specific molecular contexts can influence radiosensitivity and should guide individualized radiation dose strategies for metastatic lesions.

## Figures and Tables

**Figure 1 ijms-26-11837-f001:**
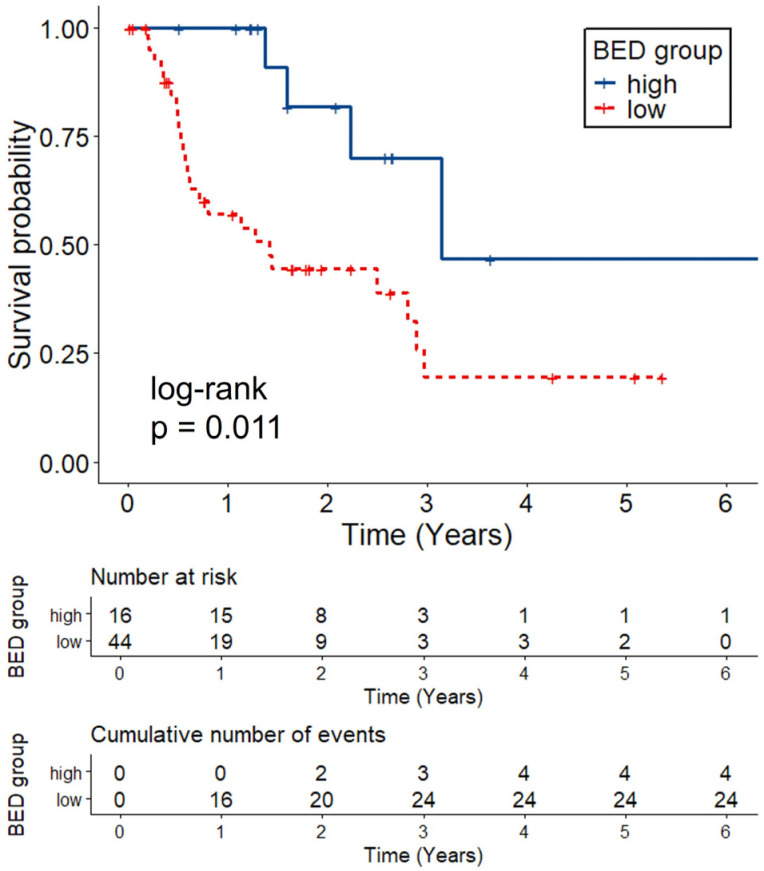
Kaplan–Meier curves of LPFS by BED group. Patients receiving high BED (≥88 Gy) showed significantly longer LPFS than those with low BED (<88 Gy) (log-rank *p* = 0.011). Abbreviation: LPFS, local progression free survival; BED, biologically effective dose.

**Figure 2 ijms-26-11837-f002:**
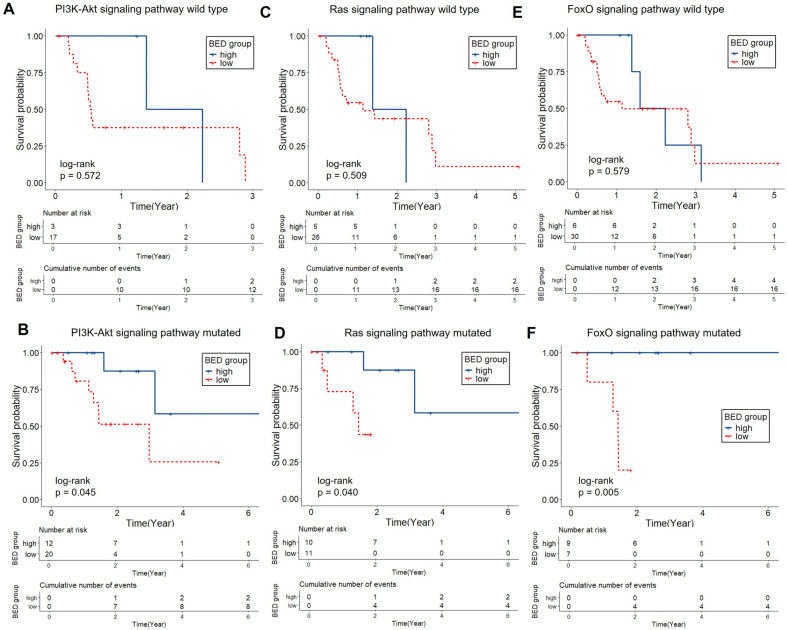
Kaplan–Meier curves of LPFS by BED group with or without PI3K-Akt, Ras, and FoxO pathway mutations. High BED was associated with significantly longer LPFS in pathway mutated group (PI3K-Akt [*n* = 32]: *p* = 0.045; Ras [*n* = 21]: *p* = 0.04; FoxO [*n* = 16]: *p* = 0.0046, (**B**,**D**,**F**)), whereas no significant difference was observed in wild-type groups (PI3K-Akt [*n* = 20]; Ras [*n* = 31]; FoxO [*n* = 36]; (**A**,**C**,**E**)). Abbreviation: BED, biologically effective dose; LPFS, local progression free survival.

**Figure 3 ijms-26-11837-f003:**
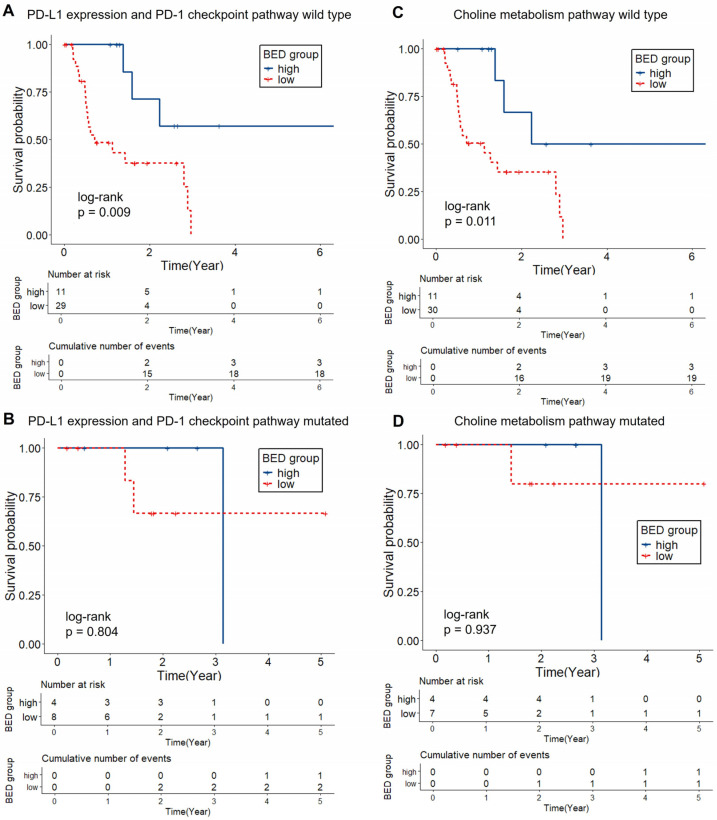
Kaplan–Meier curves of LPFS by BED group with or without PD-L1 expression and PD-1 checkpoint pathway and choline metabolism pathway mutations. High BED was associated with significantly longer LPFS in wild-type group (PD-L1 expression and PD-1 checkpoint pathway [*n* = 40]: *p* = 0.0091; choline metabolism pathway [*n* = 41]: *p* = 0.011, (**A**,**C**)), whereas no significant difference was observed in pathway mutated groups (PD-L1 expression and PD-1 checkpoint pathway [*n* = 12]; choline metabolism pathway [*n* = 11]; (**B**,**D**)). Abbreviation: BED, biologically effective dose; LPFS, local progression free survival.

**Table 1 ijms-26-11837-t001:** Characteristics of patients and metastatic lesions.

Variable	All Patients (*n* = 44)	Variable	All Lesions (*n* = 60)
Age, y, mean ± SD	54.2 ± 9.7	Target, *n* (%)	
Histology, *n* (%)		Bone	41 (68.3)
ductal	33 (75.0)	Brain	9 (15.0)
lobular	4 (9.1)	Lymph nodes	4 (6.7)
metaplastic	1 (2.3)	Breast	2 (3.3)
mixed	2 (4.6)	Bladder	1 (1.7)
mucinous	2 (4.5)	Lung	1 (1.7)
unknown	2 (4.5)	Muscle	1 (1.7)
Grade, *n* (%)		Skin	1 (1.7)
1	1 (2.3)	BED, α/β = 3, median [IQR]	75.0 [60.0, 90.0]
2	22 (50.0)	Concordance between biopsied and irradiated sites, *n* (%)	
3	17 (38.6)	Concordant	19 (31.7)
unknown	4 (9.1)	Discordant	41 (68.3)
Estrogen receptor, *n* (%)		Time interval between biopsy and radiotherapy, days, median [IQR]	154.5 [37.0, 401.0]
positive	35 (79.5)	Systemic therapy type during radiotherapy, *n* (%)	
negative	9 (20.5)	Chemotherapy ± Endocrine therapy	17 (28.3)
Progesterone receptor, *n* (%)		Targeted therapy ± Endocrine therapy	10 (16.7)
positive	19 (43.2)	HER2-directed therapy ± Endocrine therapy	3 (5.0)
negative	25 (56.8)	Endocrine therapy alone	11 (18.3)
HER2, *n* (%)		RT alone	19 (31.7)
positive	5 (11.4)		
negative	39 (88.6)		
Ki-67, %, mean ± SD	30.5 ± 23.2		
Subtype, *n* (%)			
Luminal A	11 (25.0)		
Luminal B	21 (47.7)		
Triple Negative	12 (27.3)		
NGS target lesion, *n* (%)			
Breast	16 (36.4)		
Bone	8 (18.2)		
Lymph node	6 (13.6)		
Lung	5 (11.4)		
Liver	5 (11.4)		
Others	4 (9.0)		

Abbreviation: BED, biologically effective dose; NGS, next-generation sequencing.

**Table 2 ijms-26-11837-t002:** Univariate and multivariate Cox regression analyses of local progression-free survival according to gene mutations.

Variable	Univariate			Multivariate		
	HR	95% Cl	*p*-Value	HR	95% Cl	*p*-Value
TP53	0.91	0.42–1.98	0.815	-		
PIK3CA	0.40	0.17–0.95	0.037	0.50	0.16–1.53	0.223
FAT1	1.43	0.68–3.03	0.349	-		
ZFHX3	0.77	0.34–1.76	0.539	-		
SPEN	0.46	0.17–1.26	0.130	-		
SPTA1	0.65	0.26–1.63	0.359	-		
RECQL4	1.32	0.56–3.13	0.529	-		
PRKDC	1.32	0.55–3.14	0.535	-		
ARID1B	1.23	0.5–3.04	0.655	-		
MDC1	1.17	0.47–2.91	0.737	-		
NTRK3	2.54	1.02–6.33	0.045	2.46	0.75–8.12	0.140
MYCN	0.29	0.07–1.25	0.085	0.64	0.09–4.39	0.653
ERBB3	2.42	0.89–6.55	0.083	3.42	1.12–10.46	0.031
BARD1	0.17	0.02–1.27	0.085	0.78	0.09–6.91	0.826

Abbreviation: CI, Confidence interval; HR, Hazard ratio.

**Table 3 ijms-26-11837-t003:** Univariate and multivariate Cox regression analyses of local progression-free survival according to pathway mutations.

Variable	Univariate			Multivariate		
	HR	95% Cl	*p*-Value	HR	95% Cl	*p*-Value
PI3K-Akt signaling pathway	0.23	0.1–0.56	0.001	0.18	0.04–0.87	0.033
EGFR tyrosine kinase inhibitor resistance	0.35	0.15–0.84	0.019	5.79	0.64–52.26	0.118
Ras signaling pathway	0.38	0.15–0.97	0.043	1.32	0.01–185.78	0.913
ErbB signaling pathway	0.19	0.06–0.68	0.010	0.36	0.01–15.90	0.599
FoxO signaling pathway	0.36	0.12–1.06	0.064	0.12	0.00–12.51	0.370
HIF-1 signaling pathway	0.13	0.03–0.58	0.007	0.04	0.00–2.19	0.115
VEGF signaling pathway	0.15	0.03–0.66	0.012	-		
B cell receptor signaling pathway	0.06	0.01–0.46	0.007	-		
PD-L1 expression and PD-1 checkpoint pathway in cancer	0.33	0.1–1.11	0.073	0.24	0.00–44.28	0.593
Choline metabolism in cancer	0.20	0.05–0.88	0.033	0.61	0.00–2770	0.910
Estrogen signaling pathway	0.12	0.02–0.88	0.037	0.71	0.00–3433	0.937
Notch signaling pathway	0.10	0.01–0.81	0.031	-		

Abbreviation: CI, Confidence interval; HR, Hazard ratio.

## Data Availability

The datasets are not readily available because they contain patient-level genomic data protected under institutional privacy regulations. If someone wants to request the data/or have queries from this study, contact the corresponding author (In Ah Kim, e-mail: inah228@snu.ac.kr).

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
