# Peer review of "Association Between Genomic Features and Radiation Response in Metastatic Breast Cancer Patients Undergoing Palliative Radiotherapy"

_ijms, 2025, doi:10.3390/ijms262411837_

Round 1

Reviewer 1 Report

Comments and Suggestions for Authors

Thank you for the opportunity to review your manuscript. The topic is important, and linking genomic features with radiotherapy outcomes in metastatic breast cancer is clinically meaningful. The work is interesting and offers useful exploratory observations. However, a few key issues need clarification before the manuscript can be considered further.

Major Points

  1. Small cohort and model stability
    The sample size is quite limited compared with the number of genomic variables analyzed. Some hazard ratios have very wide confidence intervals, suggesting model instability. Please highlight this limitation clearly and, if possible, simplify or reduce the multivariable models.
  2. NGS–RT site discordance
    Only a minority of biopsies matched the irradiated lesion. This may influence the genomic–LPFS associations. Please discuss how this may affect interpretation and indicate the typical time interval between biopsy and radiotherapy.
  3. Pathway alteration definition
    All mutation types (activating, LOF, VUS, CNVs) are grouped. Since these may have different biological effects, please clarify this approach or acknowledge the limitation.
  4. BED cutoff
    The 88 Gy cutoff was statistically derived; sensitivity to sample size should be discussed. Please mention this limitation or provide a brief justification.
  5. Role of systemic therapy
    Concurrent systemic treatments may influence local control. A summary of systemic therapies during radiotherapy would be helpful.

Minor Points

  • Please add the number of LPFS events.
  • Include “number at risk” in Kaplan–Meier plots.
Comments on the Quality of English Language

A brief language and formatting check is recommended.

Author Response

Comment 1: Small cohort and model stability
The sample size is quite limited compared with the number of genomic variables analyzed. Some hazard ratios have very wide confidence intervals, suggesting model instability. Please highlight this limitation clearly and, if possible, simplify or reduce the multivariable models.

Response 1: Thank you for pointing out this critical issue. We agree that the limited sample size relative to the number of genomic variables may lead to model instability, as reflected in the wide confidence intervals. As suggested, we have explicitly highlighted this limitation in the Discussion section. Please refer to the Discussion section, page 9, line 214.

Comment 2: NGS–RT site discordance
Only a minority of biopsies matched the irradiated lesion. This may influence the genomic–LPFS associations. Please discuss how this may affect interpretation and indicate the typical time interval between biopsy and radiotherapy.

Response 2: We appreciate this valid concern. We acknowledge that the spatial and temporal discordance between the biopsy and the irradiated site can introduce bias due to intratumoral heterogeneity and clonal evolution. We calculated the time interval and found the median time between biopsy and radiotherapy to be 154 days. We have added a discussion on how this discordance limits the direct interpretation of our findings in the Discussion section. Please refer to the Discussion section, page 9, line 227.

Comment 3: Pathway alteration definition

All mutation types (activating, LOF, VUS, CNVs) are grouped. Since these may have different biological effects, please clarify this approach or acknowledge the limitation.

Response 3: We agree that grouping different mutation types may obscure specific biological effects. Due to the limited sample size, we were unable to perform separate analyses for each alteration type. We have acknowledged this as a limitation in the Discussion section to prevent overinterpretation of the biological associations. Please refer to the Discussion section, page 9, line 229.

Comment 4: BED cutoff

The 88 Gy cutoff was statistically derived; sensitivity to sample size should be discussed. Please mention this limitation or provide a brief justification.

Response 4: Thank you for this comment. We recognize that the BED cutoff of 88 Gy was derived post-hoc from our specific cohort and may be sensitive to the sample size. We have added a statement in the Discussion section emphasizing that this threshold requires validation in external datasets and should be interpreted with caution. Please refer to the Discussion section, page 9, line 215.

Comment 5: Role of systemic therapy
Concurrent systemic treatments may influence local control. A summary of systemic therapies during radiotherapy would be helpful.

Response 5: We appreciate this suggestion. We agree that concurrent systemic therapy is a potential confounder for local control. We have reviewed the clinical data and added a summary of the systemic therapies administered during radiotherapy to the Table 1.

Coment 6: Please add the number of LPFS events.
Include “number at risk” in Kaplan–Meier plots.

Response 6: We appreciate this helpful suggestion to improve the clarity of the survival analysis. We have revised the Kaplan-Meier plots (Figure 1,2,3) to include both the number at risk table and the cumulative number of events table below the x-axis.

Reviewer 2 Report

Comments and Suggestions for Authors

In this manuscript, this study highlights the potential of combining radiotherapy dose parameters with genomic pathway alterations to guide precision radiation therapy in metastatic breast cancer. Generally speaking, this manuscript is well-written and organized, and the provided data can support their claims. Hence, a minor revision is recommended before publication. Here are detailed comments:

  1. The manuscript identifies 88 Gy as the optimal BED cutoff using maximally selected rank statistics; however, the clinical rationale and biological interpretability of this threshold are not sufficiently discussed. Please expand on why 88 Gy is biologically meaningful or clinically actionable, and compare it with commonly used palliative RT dose regimens.
  2. Although systemic therapy during radiotherapy was acknowledged as a limitation, the manuscript does not provide detailed information on concurrent or recent systemic treatments (e.g., CDK4/6 inhibitors, PI3K inhibitors, immunotherapy). Given that many of these agents could interact with RT response or with pathways identified (e.g., PI3K-Akt, PD-1/PD-L1), please include a table or subgroup description of systemic treatments received around the RT period, and discuss how these could confound gene-radiation interactions.
  3. Figures 2 and 3 contain multiple Kaplan-Meier curves, but sample sizes for each subgroup are very small and not shown explicitly. Please add number-at-risk tables, event counts, and sample size per subgroup directly into the figure panels or captions. Additionally, consider providing interaction p-values to statistically support the claim that BED benefit differs by pathway mutation status.
  4. Please provide more detailed information on lesion-level heterogeneity and biopsy-irradiation mismatch.

Author Response

Comment1 : The manuscript identifies 88 Gy as the optimal BED cutoff using maximally selected rank statistics; however, the clinical rationale and biological interpretability of this threshold are not sufficiently discussed. Please expand on why 88 Gy is biologically meaningful or clinically actionable, and compare it with commonly used palliative RT dose regimens.

Response1 : We appreciate this comment. Although the 88 Gy cutoff was statistically derived, it aligns well with the radiobiological distinction between palliative and locally effective dose levels when modeled with an α/β ratio of 3 Gy. Commonly used palliative regimens such as 30 Gy in 10 fractions (BED = 60 Gy) and 20 Gy in 5 fractions (BED = 46.7 Gy) fall well below this threshold. In contrast, a BED of 88 Gy corresponds to approximately 50–54 Gy in 2-Gy fractions, a dose range associated with more durable local control. Thus, this cutoff likely marks a transition from short-term palliation to dose intensities capable of overcoming intrinsic or mutation-related radioresistance. We have added a paragraph to the Discussion to clarify this rationale. Please refer to the Discussion section, page 9, line 179.

Comment 2: Although systemic therapy during radiotherapy was acknowledged as a limitation, the manuscript does not provide detailed information on concurrent or recent systemic treatments (e.g., CDK4/6 inhibitors, PI3K inhibitors, immunotherapy). Given that many of these agents could interact with RT response or with pathways identified (e.g., PI3K-Akt, PD-1/PD-L1), please include a table or subgroup description of systemic treatments received around the RT period, and discuss how these could confound gene-radiation interactions.

Response 2: Thank you for the suggestion. We reviewed systemic therapies administered during the RT period and identified 8 lesions treated with concurrent CDK4/6 inhibitors and 2 lesions with mTOR inhibitors. These details have been added to Table 1. Due to the small size of these subgroups, formal interaction or subgroup analyses were not feasible. As noted in the revised Discussion, these agents may act as potential confounders given their relevance to pathways such as PI3K–Akt–mTOR and PD-1/PD-L1, but their low frequency makes it unlikely that they account for the overall associations observed. Please refer to the Result section, page 3, Table 1 and Discussion section, page 9, line 219.

Comment 3: Figures 2 and 3 contain multiple Kaplan-Meier curves, but sample sizes for each subgroup are very small and not shown explicitly. Please add number-at-risk tables, event counts, and sample size per subgroup directly into the figure panels or captions. Additionally, consider providing interaction p-values to statistically support the claim that BED benefit differs by pathway mutation status.

Response 3: We appreciate this suggestion. We performed Cox proportional hazards models including an interaction term between BED and pathway mutation status for all analyzed pathways. Specifically, the interaction p-values were 0.406 for PI3K–Akt, 0.251 for Ras, 0.251 for choline metabolism, and 0.326 for the PD-L1 signaling pathway. Notably, for the FoxO signaling pathway, the interaction model failed to converge due to the sparsity of events within the subgroup, highlighting the statistical limitations imposed by the small sample size (n = 52 lesions). While these interaction terms did not reach statistical significance (all p > 0.05), we believe these results reflect limited statistical power rather than a definitive absence of differential BED effects. Consequently, we elected not to display interaction p-values in Figures 2 and 3 to avoid potential misinterpretation of the findings as 'negative' solely due to insufficient power. Instead, we describe these trends qualitatively in the text and have explicitly acknowledged in the Limitations section that the study was underpowered for formal interaction testing. As requested, number-at-risk and cumulative number of events tables have been added directly to Figures 2 and 3, and sample sizes for each subgroup have been explicitly stated in the figure captions to enhance clarity and transparency.

Comment 4: Please provide more detailed information on lesion-level heterogeneity and biopsy-irradiation mismatch.

Response 4: We agree that lesion-level heterogeneity and biopsy–irradiation mismatch are important considerations. We have added the median time interval between biopsy and radiotherapy (154 days) to the Results section. Furthermore, we revised the Discussion to explicitly acknowledge that frequent spatial discordance and this temporal separation may introduce clonal evolution and inter-lesional heterogeneity, limiting the direct correlation between genomic alterations and radiation response. Please refer to the Discussion section, page 9, line 227.

Reviewer 3 Report

Comments and Suggestions for Authors

In the present manuscript Choi et al. examine the mutational determinants that govern the correlation between the biologically effective radiation dose and local progression free survival in metastatic breast cancer. The authors uncover signaling pathways, namely, PI3K, RAS and FoxO in which high BED correlates with better LPFS. Conversely, PD-L1/PD checkpoint and Choline metabolism pathways offer better BED-LPFS correlations when they are unaltered. These observation are likely to serve as biomarkers informing outcomes during palliative radiation therapy. I only have a few minor comments:

1) In Table 2, aside from ERBB3 mutations being a predictor of shorter LPFS, the authors also observe BARD1 as being a similar predictor. Given, the role of BARD1 in Homologous recombination, one would expect BARD1 mutant tumors to fare better in terms of LPFS. What is the likely explanation for this? Is this due to a basal increase in PD-L1 expression which increases immune evasion? 

2) PIK3CA and RAS are often co-mutated in a variety of cancers. How many instances of this are observed? Does PIK3CA-RAS pathway co-alteration further increase the correlation between BED and LPFS?

3) Since the RAS and PI3K/AKT pathways are interlinked, is there any additional data on downstream MAPK effectors or is the BED/LPFS correlation on based on alterations of RAS and upstream factors? 

Author Response

Comment1 : In Table 2, aside from ERBB3 mutations being a predictor of shorter LPFS, the authors also observe BARD1 as being a similar predictor. Given, the role of BARD1 in Homologous recombination, one would expect BARD1 mutant tumors to fare better in terms of LPFS. What is the likely explanation for this? Is this due to a basal increase in PD-L1 expression which increases immune evasion? 

Response 1: Thank you for this comment. In our dataset, BARD1 mutations were identified in only three patients (corresponding to six lesions). The small number of mutation-positive cases and the limited number of progression events in this subgroup resulted in unstable hazard ratio estimates with wide confidence intervals. Therefore, the apparent direction of association in Table 2 likely reflects statistical noise rather than a reproducible biological signal. Under these conditions, our data do not support a mechanistic interpretation for BARD1, nor do they suggest a role for PD-L1–mediated immune evasion. We consider the observed pattern to be an artifact of the small sample size.

Comment 2: PIK3CA and RAS are often co-mutated in a variety of cancers. How many instances of this are observed? Does PIK3CA-RAS pathway co-alteration further increase the correlation between BED and LPFS?

Response 2: We appreciate the reviewer’s insightful comment regarding the potential interaction between PIK3CA and RAS pathway alterations. In our cohort, PI3K–Akt pathway alterations were identified in 32 lesions and Ras pathway alterations in 21 lesions; among these, 19 lesions exhibited concurrent co-alteration. To evaluate whether this co-alteration modified the association between BED and LPFS, we performed an interaction analysis using a Cox proportional hazards model. The interaction term was not statistically significant (p = 0.442), indicating that the co-alteration did not statistically significantly modify the BED–LPFS relationship in this dataset. However, given the limited sample size and number of events, we acknowledge that the analysis may have been underpowered to detect synergistic effects. Therefore, we have interpreted these findings with caution.

Comment 3: Since the RAS and PI3K/AKT pathways are interlinked, is there any additional data on downstream MAPK effectors or is the BED/LPFS correlation on based on alterations of RAS and upstream factors?
Response 3: We appreciate the reviewer’s thoughtful comment regarding upstream versus downstream signaling within the RAS–PI3K/AKT–MAPK axis. Although the TruSight Oncology 500 panel includes key downstream MAPK effectors such as MAPK1 (ERK2) and MAPK3 (ERK1), no pathogenic mutations in these genes were detected in our cohort. Other MAPK-related genes covered by the panel also exhibited mutation frequencies too low to permit meaningful survival analysis.
Because functional MAPK activation often depends on post-translational modifications such as ERK phosphorylation, which cannot be assessed using DNA-based sequencing, the BED–LPFS association observed in our study reflects upstream genomic alterations (RAS or PI3K/AKT) rather than downstream effector events. We have clarified this distinction and acknowledged the related methodological limitations in the revised Discussion. Please refer to the Discussion section, page 9, line 198.